# A Unified Framework for Convolution-based Graph Neural Networks

## Abstract

Graph Convolutional Networks (GCNs) have attracted a lot of research interest in the machine learning community in recent years. Although many variants have been proposed, we still lack a systematic view of different GCN models and deep understanding of the relations among them. In this paper, we take a step forward to establish a unified framework for convolution-based graph neural networks, by formulating the basic graph convolution operation as an optimization problem in the graph Fourier space. Under this framework, a variety of popular GCN models, including the vanilla-GCNs, attention-based GCNs and topology-based GCNs, can be interpreted as a same optimization problem but with different carefully designed regularizers. This novel perspective enables a better understanding of the similarities and differences among many widely used GCNs, and may inspire new approaches for designing better models. As a showcase, we also present a novel regularization technique under the proposed framework to tackle the oversmoothing problem in graph convolution. The effectiveness of the newly designed model is validated empirically.

## 1 Introduction

Recent years have witnessed a fast development in graph processing by generalizing convolution operation to graph-structured data, which is known as Graph Convolutional Networks (GCNs) (Kipf & Welling, 2017). Due to the great success, numerous variants of GCNs have been developed and extensively adopted in the field of social network analysis (Hamilton et al., 2017; Wu et al., 2019a; Veličković et al., 2018), biology (Zitnik et al., 2018), transportation forecasting (Li et al., 2017) and natural language processing (Wu et al., 2019b; Yao et al., 2019).

Inspired by GCN, a wide variety of convolution-based graph learning approaches are proposed to enhance the generalization performance of graph neural networks. Several research aim to achieve higher expressiveness by exploring higher-order information or introducing additional learning mechanisms like attention modules. Although proposed from different perspectives, their exist some connections between these approaches. For example, attention-based GCNs like GAT (Veličković et al., 2018) and AGNN (Thekumparampil et al., 2018) share the similar intention by adjusting the adjacency matrix with a function of edge and node features. Similarly, TAGCN (Du et al., 2017) and MixHop (Kapoor et al., 2019) can be viewed as particular instances of PPNP (Klicpera et al., 2018) under certain approximation. However, the relations among these graph learning models are rarely studied and the comparisons are still limited in analyzing generalization performances on public datasets. As a consequence, we still lack a systematic view of different GCN models and deep understanding of the relations among them.

In this paper, we resort to the techniques in graph signal processing and attempt to understand GCN-based approaches from a general perspective. Specifically, we present a unified graph convolution framework by establishing graph convolution operations with optimization problems in the graph Fourier domain. We consider a Laplacian regularized least squares optimization problem and show that most of the convolution-based approaches can be interpreted in this framework by adding carefully designed regularizers. Besides vanilla GCNs, we also extend our framework to formulating non-convolutional operations (Xu et al., 2018a; Hamilton et al., 2017), attention-based GCNs (Veličković et al., 2018; Thekumparampil et al., 2018) and topology-based GCNs (Klicpera et al., 2018; Kapoor et al., 2019), which cover a large fraction of the state-of-the-art graph learning ap-

proaches. This novel perspective provides a re-interpretation of graph convolution operations and enables a better understanding of the similarities and differences among many widely used GCNs, and may inspire new approaches for designing better models.

As a conclusion, we summarize our contributions as follow:

1. We introduce a unified framework for convolution-based graph neural networks and interpret various convolution filters as carefully designed regularizers in the graph Fourier domain, which provides a general methodology for evaluating and relating different graph learning modules.

2. Based on the proposed framework, we provide new insights on understanding the limitations of GCNs and show new directions to tackle common problems and improve the generalization performance of current graph neural networks in the graph Fourier domain. Additionally, the unified framework can serve as a once-for-all platform for expert-designed modules on convolution-based approaches, where newly designed modules can be easily implemented on other networks as a plug-in module with trivial adaptations. We believe that our framework can provide convenience for designing new graph learning modules and searching for better combinations.

3. As a showcase, we present a novel regularization technique under the proposed framework to alleviate the oversmoothing problem in graph representation learning. As shown in Section 4, the newly designed regularizer can be implemented on several convolution-based networks and effectively improve the generalization performance of graph learning models.

## 2 PRELIMINARY

We start with an overview of the basic concepts of graph signal processing. Let $\mathcal{G} = (\mathbb{V}, \boldsymbol{A})$ denote a graph with node feature vectors where $\mathbb{V}$ represents the vertex set consisting of nodes $\{\boldsymbol{v}_1, \boldsymbol{v}_2, \ldots, \boldsymbol{v}_N\}$ and $\boldsymbol{A} = (a_{ij}) \in \mathbb{R}^{N \times N}$ is the adjacency matrix implying the connectivity between nodes in the graph. Let $\boldsymbol{D} = \text{diag}(d(1), \ldots, d(N)) \in \mathbb{R}^{N \times N}$ be the degree matrix of $\boldsymbol{A}$ where $d(i) = \sum_{j \in \mathbb{V}} a_{ij}$ is the degree of vertex $i$. Then, $\boldsymbol{L} = \boldsymbol{D} - \boldsymbol{A}$ is the combinatorial Laplacian and $\tilde{\boldsymbol{L}} = \boldsymbol{I} - \boldsymbol{D}^{(-1/2)} \boldsymbol{A} \boldsymbol{D}^{(-1/2)}$ is the normalized Laplacian of $\mathcal{G}$. Additionally, we let $\tilde{\boldsymbol{A}} = \boldsymbol{A} + \boldsymbol{I}$ and $\tilde{\boldsymbol{D}} = \boldsymbol{D} + \boldsymbol{I}$ denote the augmented adjacency and degree matrices with added self-loops. Then $\tilde{\mathcal{L}}_{\text{sym}} = \boldsymbol{I} - \tilde{\boldsymbol{D}}^{-1/2} \tilde{\boldsymbol{A}} \tilde{\boldsymbol{D}}^{-1/2}$ ($\tilde{\mathcal{A}}_{\text{sym}} = \tilde{\boldsymbol{D}}^{-1/2} \tilde{\boldsymbol{A}} \tilde{\boldsymbol{D}}^{-1/2}$) and $\tilde{\mathcal{L}}_{\text{rw}} = \boldsymbol{I} - \tilde{\boldsymbol{D}}^{-1} \tilde{\boldsymbol{A}}$ ($\tilde{\mathcal{A}}_{\text{rw}} = \tilde{\boldsymbol{D}}^{-1} \tilde{\boldsymbol{A}}$) are the augmented symmetric normalized and random walk normalized Laplacian (augmented adjacency matrices) of $\mathcal{G}$, respectively.

Let $\boldsymbol{x} \in \mathbb{R}^N$ be a signal on the vertices of the graph. The spectral convolution is defined as a function of a filter $g_\theta$ parameterized in the Fourier domain (Kipf & Welling, 2017):

$$g_\theta \star \boldsymbol{x} = \boldsymbol{U} g_\theta(\Lambda) \boldsymbol{U}^T \boldsymbol{x}, \tag{1}$$

where $\boldsymbol{U}$ and $\Lambda$ are the eigenvectors and eigenvalues of the normalized Laplacian $\tilde{\boldsymbol{L}}$. Also, we follow Hoang & Maehara (2019) and define the variation $\Delta$ and $\tilde{\boldsymbol{D}}$-inner product as:

$$\Delta(\boldsymbol{x}) = \sum_{i,j \in \mathbb{V}} a_{ij}(\boldsymbol{x}(i) - \boldsymbol{x}(j))^2 = \boldsymbol{x}^T \boldsymbol{L} \boldsymbol{x}, \quad (\boldsymbol{x}, \boldsymbol{y})_{\tilde{\boldsymbol{D}}} = \sum_{i \in \mathbb{V}}(d(i) + 1)\boldsymbol{x}(i)\boldsymbol{y}(i) = \boldsymbol{x}^T \tilde{\boldsymbol{D}} \boldsymbol{y}, \tag{2}$$

which specifies the smoothness and importance of the signal respectively.

## 3 UNIFIED GRAPH CONVOLUTION FRAMEWORK

With the success of GCNs, a wide variety of convolution-based approaches are proposed which progressively enhance the expressive power and generalization performance of graph neural networks. Despite the effectiveness of GCN and its derivatives on specific tasks, there still lack a comprehensive understanding on the relations and differences among various graph learning modules.

Graph signal processing is a powerful technique which has been adopted in several graph learning researches (Kipf & Welling, 2017; Hoang & Maehara, 2019; Zhao & Akoglu, 2019). However, existing researches mainly focus on analyzing the properties of GCNs while ignore the connections between different graph learning modules. Innovatively, in this work, we consider interpreting convolution-based approaches from a general perspective with graph signal processing techniques.

In specific, we establish the connections between graph convolution operations and optimization problems in graph Fourier space, showing the effect of each module explicitly with specific regularizers. This novel perspective provides a systematic view of different GCN models and deep understanding of the relations among them.

## 3.1 Unified Graph Convolution Framework

Several researches have proved that, in the field of graph signal processing, the representative features are mostly preserved in the low-frequency signals while noises are mostly contained in the high-frequency signals (Hoang & Maehara, 2019). Based on this observation, numerous graph representation learning methods are designed to decrease the high-frequency components, which can be viewed as low-pass filters in the graph Fourier space. With similar inspiration, we consider a Laplacian regularized least squares optimization problem with graph signal regularizers and attempt to build connections with these filters.

**Definition 1 Unified Graph Convolution Framework.** *Graph convolution filters can be achieved by solving the following Laplacian regularized least squares optimization:*

$$\min_{\bar{\boldsymbol{X}}} \ \sum_{i \in \mathbb{V}} \|\bar{\boldsymbol{x}}(i) - \boldsymbol{x}(i)\|_{\tilde{\boldsymbol{D}}}^2 + \lambda \mathcal{L}_{\mathrm{reg}}, \tag{3}$$

*where* $\|\boldsymbol{x}\|_{\tilde{\boldsymbol{D}}} = \sqrt{(\boldsymbol{x}, \boldsymbol{x})_{\tilde{\boldsymbol{D}}}}$ *denotes the norm induced by* $\tilde{\boldsymbol{D}}$.

In the following sections, we will show that a wide range of convolution-based graph neural networks can be derived from Definition 1 with different carefully designed regularizers, and provide new insights on understanding different graph learning modules from the graph signal perspective.

### 3.1.1 Graph convolutional networks

Graph convolutional networks (GCNs) (Kipf & Welling, 2017) are the foundation of numerous graph learning models and have received widespread concerns. Several researches have demonstrated that the vanilla GCN is essentially a type of Laplacian smoothing over the whole graph, which makes the features of the connected nodes similar. Therefore, to reformulate GCNs in the graph Fourier space, we consider utilizing the variation $\Delta(\boldsymbol{x})$ as the regularizer.

**Definition 2 Vanilla GCNs.** *Let* $\bar{\boldsymbol{x}}(i)_{i \in \mathbb{V}}$ *be the estimation of the input observation* $\boldsymbol{x}(i)_{i \in \mathbb{V}}$. *A low-pass filter:*

$$\bar{\boldsymbol{X}} = \tilde{\mathcal{A}}_{\mathrm{rw}} \boldsymbol{X}, \tag{4}$$

*is the first-order approximation of the optimal solution of the following optimization:*

$$\min_{\bar{\boldsymbol{X}}} \ \sum_{i \in \mathbb{V}} \|\bar{\boldsymbol{x}}(i) - \boldsymbol{x}(i)\|_{\tilde{\boldsymbol{D}}}^2 + \sum_{i,j \in \mathbb{V}} a_{ij} \|\bar{\boldsymbol{x}}(i) - \bar{\boldsymbol{x}}(j)\|_2^2. \tag{5}$$

*Derivations of the definitions are presented in Appendix A.*

As the eigenvalues of the approximated filter $\tilde{\mathcal{A}}_{\mathrm{rw}}$ are bounded by 1, it resembles a low-pass filter that removes the high-frequency signals. By exchanging $\tilde{\mathcal{A}}_{\mathrm{rw}}$ with $\tilde{\mathcal{A}}_{\mathrm{sym}}$ (which has the same eigenvalues as $\tilde{\mathcal{A}}_{\mathrm{rw}}$), we obtain the same formulation adopted in GCNs.

It has been stated that the second term $\Delta(\boldsymbol{x})$ in Eq.(5) measures the variation of the estimation $\bar{\boldsymbol{x}}$ over the graph structure. By adding this regularizer to the objective function, the obtained filter emphasizes the low-frequency signals through minimizing the variation over the local graph structure, while keeping the estimation close to the input in the graph Fourier space.

### 3.1.2 Non-convolutional Operations

**Residual Connection.** Residual connection is first proposed by He et al. (2016) and has been widely adopted in graph representation learning approaches. In the vanilla GCNs, norms of the eigenvalues of the filter $\tilde{\mathcal{A}}_{\mathrm{rw}}$ (or $\tilde{\mathcal{A}}_{\mathrm{sym}}$) are bounded by 1 which ensures numerical stability in the training procedure. However, on the other hand, signals in all frequency band will shrink as the convolution layer stacks, leading to a consistent information loss. Therefore, adding the residual connection is deemed to preserve the strength of the input signal.

**Definition 3 Residual Connection.** *A graph convolution filter with residual connection:*

$$\bar{\boldsymbol{X}} = \tilde{\mathcal{A}}_{\mathrm{rw}} \boldsymbol{X} + \epsilon \boldsymbol{X}, \tag{6}$$

*where $\epsilon > 0$ controls the strength of residual connection, **is the first-order approximation of** the optimal solution of the following optimization:*

$$\min_{\bar{\boldsymbol{X}}} \sum_{i \in \mathbb{V}} (\|\bar{\boldsymbol{x}}(i) - \boldsymbol{x}(i)\|_{\tilde{\boldsymbol{D}}}^2 - \epsilon \|\bar{\boldsymbol{x}}(i)\|_{\tilde{\boldsymbol{D}}}^2) + \sum_{i,j \in \mathbb{V}} a_{ij} \|\bar{\boldsymbol{x}}(i) - \bar{\boldsymbol{x}}(j)\|_2^2. \tag{7}$$

By adding the negative regularizer to penalize the estimations with small norms, we can induce the same formulation as the vanilla graph convolution with residual connection.

**Concatenation.** Concatenation is practically a residual connection with different learning weights.
**Definition 3' Concatenation.** *A graph convolution filter concatenating with the input signal:*

$$\bar{\boldsymbol{X}} = \tilde{\mathcal{A}}_{\mathrm{rw}} \boldsymbol{X} + \epsilon \boldsymbol{X} \Theta \Theta^T, \tag{8}$$

*is the first-order approximation of the optimal solution of the following optimization:*

$$\min_{\bar{\boldsymbol{X}}} \sum_{i \in \mathbb{V}} (\|\bar{\boldsymbol{x}}(i) - \boldsymbol{x}(i)\|_{\tilde{\boldsymbol{D}}}^2 - \epsilon \|\bar{\boldsymbol{x}}(i)\Theta\|_{\tilde{\boldsymbol{D}}}^2) + \sum_{i,j \in \mathbb{V}} a_{ij} \|\bar{\boldsymbol{x}}(i) - \bar{\boldsymbol{x}}(j)\|_2^2, \tag{9}$$

*where $\epsilon > 0$ controls the strength of concatenation and $\Theta$ is the learning coefficient.*

Although the learning weights $\Theta\Theta^T$ has a constrained expressive capability, it can be compensated by the following feature learning modules.

### 3.1.3 ATTENTION-BASED CONVOLUTIONAL NETWORKS

Since the convolution filters in GCNs are dependent only on the graph structure, GCNs are proved to have restricted expressive power and may cause the oversmoothing problem. Several researches try to introduce the attention mechanism to the convolution filter, learning to assign different edge weights at each layer based on nodes and edges. GAT (Veličković et al., 2018) and AGNN (Thekumparampil et al., 2018) compute the attention coefficients as a function of the features of connected nodes, while ECC (Simonovsky & Komodakis, 2017) and GatedGCN (Bresson & Laurent, 2017) consider the activations for each connected edge. Although these approaches have different insights, they can be all formulated as (See details in Appendix A):

$$p_{ij} = a_{ij} f_\theta(\boldsymbol{x}(i), \boldsymbol{x}(j), e_{ij}), \ \ i,j \in \mathbb{V}, \tag{10}$$

where $e_{ij}$ denotes the edge representation if applicable. Therefore, we replace $a_{ij}$ in Definition 2 with learned coefficients to enforce different regularization strength on the connected edges.

**Definition 4 Attention-based GCNs.** *An attention-based graph convolution filter:*

$$\bar{\boldsymbol{X}} = \boldsymbol{P} \boldsymbol{X}, \tag{11}$$

*is the first-order approximation of the optimal solution of the following optimization:*

$$\min_{\bar{\boldsymbol{X}}} \sum_{i \in \mathbb{V}} \|\bar{\boldsymbol{x}}(i) - \boldsymbol{x}(i)\|_{\tilde{\boldsymbol{D}}}^2 + \sum_{i,j \in \mathbb{V}} p_{ij} \|\bar{\boldsymbol{x}}(i) - \bar{\boldsymbol{x}}(j)\|_2^2, \quad s.t. \sum_{j \in \mathbb{V}} p_{ij} = \tilde{\boldsymbol{D}}_{ii}, \forall i \in \mathbb{V}. \tag{12}$$

Notice that we use a *normalization trick* to constrain the degree of attention matrix to be the same as the original degree matrix $\tilde{\boldsymbol{D}}$ as we want to preserve the strength of the regularization for each node. The formulated filter $\boldsymbol{P}$ corresponds to the matrix $\tilde{\boldsymbol{D}}^{-1}\boldsymbol{p}$ with row sum equals to 1, which is also consistent with most of the attention-based approaches after normalization. Through adjusting the regularization strength for edges, nodes with higher attention coefficients tend to have similar features while the distance for nodes with low attention coefficients will be further.

### 3.1.4 TOPOLOGY-BASED CONVOLUTIONAL NETWORKS

Attention-based approaches are mostly designed based on the local structure. Besides focusing on the first-order adjacency matrix, several approaches (Klicpera et al., 2018; 2019; Kapoor et al., 2019; Du et al., 2017) propose to adopt the structural information in the multi-hop neighborhood,

which are referred to as topology-based convolutional networks. We start with an analysis of PPNP (Klicpera et al., 2018) and then derive a general formulation for topology-based approaches.

**PPNP.** PPNP provides insights towards the propagation scheme by combining message-passing function with personalized PageRank. As proved in (Xu et al., 2018b), the influence of node $i$ on node $j$ is proportional to a k-step random walk, which converges to the limit distribution with multiple stacked convolution layers. By involving the restart probability, PPNP is able to preserve the *starting node $i$*'s information. Similarly, in Definition 2, the first term can also be viewed as a regularization of preserving the original signal information. Therefore, we may achieve the same purpose by adjusting the regularization strength.

**Definition 5 PPNP.** *A graph convolution filter with personalized propagation (PPNP):*

$$\bar{X} = \alpha(I_n - (1 - \alpha)\tilde{\mathcal{A}}_{\mathrm{rw}})^{-1}X, \tag{13}$$

*is equivalent to the optimal solution of the following optimization:*

$$\min_{\bar{X}} \quad \alpha \sum_{i \in \mathbb{V}} \|\bar{x}(i) - x(i)\|_{\tilde{D}}^2 + (1 - \alpha) \sum_{i,j \in \mathbb{V}} a_{ij}\|\bar{x}(i) - \bar{x}(j)\|_2^2, \tag{14}$$

*where $\alpha \in (0, 1]$ is the restart probability.*

Higher $\alpha$ means a higher possibility to teleport back to the starting node, which is consistent with the higher regularization on the original signal in (14).

**Multi-hop PPNP.** One of the possible weakness of the original PPNP is that personalized PageRank only utilizes the regularizer over the local structure. Therefore, we may improve the expressive capability by involving multi-hop information, which is equivalent to adding regularizers for higher-order variations.

**Definition 6 Multi-hop PPNP.** *Let $t$ be the highest order adopted in the algorithm. A graph convolution filter with multi-hop personalized propagation (Multi-hop PPNP):*

$$\bar{X} = \alpha_0(I_n - \sum_{k=1}^{t} \alpha_k \tilde{\mathcal{A}}_{\mathrm{rw}}^k)^{-1}X, \tag{15}$$

*where $\sum_{k=0}^{t} \alpha_k = 1$, $\alpha_0 > 0$ and $\alpha_k \geq 0, k = 1, 2, \ldots, t$, is equivalent to the optimal solution of the following optimization:*

$$\min_{\bar{X}} \quad \alpha_0 \sum_{i \in \mathbb{V}} \|\bar{x}(i) - x(i)\|_{\tilde{D}}^2 + \sum_{k=1}^{t} \alpha_k \sum_{i,j \in \mathbb{V}} a_{ij}^{(k)}\|\bar{x}(i) - \bar{x}(j)\|_2^2, \tag{16}$$

*where $a_{ij}^{(k)}$ is proportional to the transition probability of the k-step random walk and the same normalization trick in Section 3.1.3 is adopted on $\{a_{ij}^{(k)}\}$.*

Solving Eq.(15) directly is computationally expensive. Therefore, we derive a first-order approximation by Taylor expansion and result in the form of:

$$\bar{X} = (\sum_{i=0}^{T} \alpha_i \tilde{\mathcal{A}}_{\mathrm{rw}}^i)X + O(\tilde{\mathcal{A}}_{\mathrm{rw}}^T X). \tag{17}$$

As the norm of the eigenvalues of $\tilde{\mathcal{A}}_{\mathrm{rw}}$ are bounded by 1, we can keep the first term in Eq.(17) as a close approximation.

By comparing the approximated solution with topology-based graph convolutional networks, we find that most of the approaches can be reformulated as particular instances of Definition 6. For example, the formulation for Mixhop (Kapoor et al., 2019) can be derived as an approximation of Eq.(17) if we let $t = 2$ and $\alpha_0 = \alpha_1 = \alpha_2 = 1/3$. Different learning weights can be applied to each hop as Section 3.1.2 to concatenate multi-hop signals. See more examples in Appendix B.

## 3.2 REMARKS

In this section, we build a bridge between graph convolution operations and optimization problems in the graph Fourier space and provide insights into interpreting graph convolution operations with regularizers. For conclusion, we rewrite the general form of the unified framework as follow.

**Definition 1' Unified Graph Convolution Framework.** *Convolution-based graph neural networks can be reformulated (after approximation) as particular instances of the optimal solution of the following optimization problem:*

$$\min_{\bar{\boldsymbol{X}}} \quad \alpha_0 \sum_{i \in \mathbb{V}} (\|\bar{\boldsymbol{x}}(i) - x(\boldsymbol{i})\|_{\bar{\boldsymbol{D}}}^2 - \underbrace{\epsilon \|\bar{\boldsymbol{x}}(i)\Theta\|_{\tilde{\boldsymbol{D}}}^2}_{\mathrm{Non-Conv}}) + \sum_{k=1}^{t} \alpha_k \underbrace{\sum_{i,j \in \mathbb{V}} p_{ij}^{(k)} \underbrace{\|\bar{\boldsymbol{x}}(i)\Theta^{(k)} - \bar{\boldsymbol{x}}(j)\Theta^{(k)}\|_2^2}_{\mathrm{Attention-based}}}_{\mathrm{Topology-based}} + \lambda \mathcal{L}_{\mathrm{reg}},$$

(18)

*where $\sum_{k=0}^{t} \alpha_k = 1$, $\alpha_k \geq 0$ and $\sum_{j \in \mathbb{V}} p_{ij} = \tilde{\boldsymbol{D}}_{ii}, \forall i \in \mathbb{V}$.*

If we let $d$ be the feature dimension of $\boldsymbol{X}$, then $\Theta, \Theta^{(k)} \in \mathbb{R}^{d \times d}$ are the corresponding learning weights. $\mathcal{L}_{\mathrm{reg}}$ corresponds to the personalized regularizer based on the framework, which can be effective if carefully designed as we will show in Section 4.

By establishing the unified framework, we interpret various convolution filters as carefully designed regularizers in the graph Fourier domain, which provides new insights on understanding graph learning modules from the graph signal perspective. Several graph learning modules are reformulated as smoothing regularizers over the graph structure with different intentions. While vanilla GCNs focus on minimizing the variation over the local graph structure, attention-based and topology-based GCNs take a step forward and concentrate on the differences between connected edges and graph structure with larger receptive field. This novel perspective enables a better understanding of the similarities and differences among many widely used GCNs, and may inspire new approaches for designing better models.

## 4 TACKLING OVERSMOOTHING UNDER THE UNIFIED FRAMEWORK

Based on the proposed framework, we provide new insights on understanding the limitations of GCNs and inspire a new line of work towards designing better graph learning models. As a showcase, we present a novel regularization technique under the framework to tackle the oversmoothing problem. It is shown that the newly designed regularizer can be easily implemented on other convolution-based networks with trivial adaptations and effectively improve the generalization performances of graph learning approaches.

### 4.1 REGULARIZATION ON FEATURE VARIANCE

Here, we adopt the definition of *feature-wise* oversmoothing in (Zhao & Akoglu, 2019). Due to multiple layers of Laplacian smoothing, all features fall into the same subspace spanned by the dominated eigenvectors of the normalized adjacency matrix, which also corresponds to the similar situation described in (Klicpera et al., 2018). To tackle this problem, we propose to penalize the features when they are close to each other. Specifically, we consider the pairwise distance between normalized features, which is summarized as:

$$\delta(\boldsymbol{X}) = \frac{1}{d^2} \sum_{i,j \in d} \|\boldsymbol{x}_{\cdot i}/\|\boldsymbol{x}_{\cdot i}\| - \boldsymbol{x}_{\cdot j}/\|\boldsymbol{x}_{\cdot j}\|\|_2^2,$$

(19)

where $d$ is the feature dimension and $\boldsymbol{x}_{\cdot i} \in \mathbb{R}^n$ represents the $i$-th dimension for all nodes. Therefore, Eq.(19) can be interpreted as a feature variance regularizer, representing the distance between features after normalization. By adding this regularizer to the unified framework, the proposed filter should have the property to drive different features away.

**Definition 7 Regularized Feature Variance.** *Let $\otimes$ be the Kronecker product operator, $\mathrm{vec}(\boldsymbol{X}) \in \mathbb{R}^{nd}$ be the vectorized signal $\boldsymbol{X}$. Let $\boldsymbol{D}_X$ be a diagonal matrix whose value is defined by $\boldsymbol{D}_X(i,i) = \|\boldsymbol{x}_{\cdot i}\|_2$. A graph convolution filter with regularized feature variance:*

$$\mathrm{vec}(\bar{\boldsymbol{X}}) = (\boldsymbol{I}_n \otimes [(\alpha_1 + \alpha_2)\boldsymbol{I} - \alpha_2\tilde{\mathcal{A}}_{\mathrm{rw}}] - \alpha_3[\boldsymbol{D}_x^{-1}(\boldsymbol{I} - \frac{1}{d}\boldsymbol{1}\boldsymbol{1}^T)\boldsymbol{D}_x^{-1}] \otimes \tilde{\boldsymbol{D}}^{-1})^{-1}\mathrm{vec}(\boldsymbol{X}) \quad (20)$$

*is equivalent to the optimal solution of the following optimization:*

$$\min_{\bar{\boldsymbol{X}}} \alpha_1 \sum_{i \in \mathbb{V}} \|\bar{\boldsymbol{x}}(i) - \boldsymbol{x}(i)\|_{\tilde{\boldsymbol{D}}}^2 + \alpha_2 \sum_{i,j \in \mathbb{V}} a_{ij}\|\bar{\boldsymbol{x}}(i) - \bar{\boldsymbol{x}}(j)\|_2^2 - \alpha_3 \frac{1}{d} \sum_{i,j \in d} \|\bar{\boldsymbol{x}}_{\cdot i}/\|\boldsymbol{x}_{\cdot i}\| - \bar{\boldsymbol{x}}_{\cdot j}/\|\boldsymbol{x}_{\cdot j}\|\|_2^2, \quad (21)$$

Table 1: Test accuracy (%) on transductive learning datasets. We report mean values and standard deviations in 30 independent experiments. The best results are highlighted with **boldface**.

| Method | Dataset | Citeseer | Cora | Pubmed |
|---|---|---|---|---|
| **Vanilla** | FastGCN (Chen et al., 2018) | 68.8±0.6 | 79.8±0.3 | 76.8±0.6 |
| | DGI (Veličković et al., 2019) | 71.8±0.7 | 82.3±0.6 | 76.8±0.6 |
| | GIN (Xu et al., 2018a) | 66.1±0.9 | 77.6±1.1 | 77.0±1.2 |
| | SGC (Wu et al., 2019a) | 71.9±0.1 | 81.0±0.0 | 78.9±0.0 |
| | GCN (Kipf & Welling, 2017) | 70.3±0.4 | 81.5±0.5 | 79.0±0.4 |
| | **GCN+reg (ours)** | **72.2±0.4** | **83.6±0.3** | **79.8±0.2** |
| **Attention** | AGNN (Thekumparampil et al., 2018) | 71.6±0.5 | 82.7±0.4 | 78.9±0.4 |
| | GatedGCN (Bresson & Laurent, 2017) | 72.0±0.4 | 82.4±0.6 | 78.9±0.3 |
| | MoNet (Monti et al., 2017) | - | 81.7±0.5 | 78.8±0.4 |
| | GAT (Veličković et al., 2018) | 72.5±0.7 | 83.0±0.6 | 78.5±0.3 |
| | **GAT+reg (ours)** | **73.3±0.4** | **83.9±0.6** | **80.3±0.3** |
| **Topology** | TAGCN (Du et al., 2017) | 70.9 | 82.5 | **81.1** |
| | MixHop (Kapoor et al., 2019) | 71.4±0.8 | 81.9±0.4 | 80.8±0.6 |
| | APPNP (Klicpera et al., 2018) | 70.5±0.9 | 82.7±0.8 | 79.4±0.6 |
| | **APPNP+reg (ours)** | **71.9±0.4** | **84.0±0.6** | 80.2±0.3 |

*where $\alpha_1 > 0$, $\alpha_2, \alpha_3 \geq 0$. For computation efficiency, we approximate $\|\bar{\boldsymbol{x}}_{\cdot i}\|$ with $\|\boldsymbol{x}_{\cdot i}\|$ as we assume that a single convolution filter provides little effect to the norm of features.*

Calculating the Kronecker product and inverse operators are computationally expensive. Nevertheless, we can approximate Eq.(20) via Taylor expansion with an iterative algorithm. If we let:

$$\boldsymbol{A} = (\alpha_1 + \alpha_2)\boldsymbol{I} - \alpha_2\tilde{\mathcal{A}}_{\mathrm{rw}}, \qquad \boldsymbol{B} = \boldsymbol{I}_n, \tag{22}$$

$$\boldsymbol{C} = -\alpha_3\tilde{\boldsymbol{D}}^{-1}, \qquad \boldsymbol{D} = \boldsymbol{D}_x^{-1}(1 - \frac{1}{d}\boldsymbol{1}\boldsymbol{1}^T)\boldsymbol{D}_x^{-1}. \tag{23}$$

Then, a $t$-order approximated formulation is summarized as:

$$\bar{\boldsymbol{X}}^{(0)} = \boldsymbol{X}, \tag{24}$$

$$\bar{\boldsymbol{X}}^{(k+1)} = \boldsymbol{X} + \bar{\boldsymbol{X}}^{(k)} - \boldsymbol{A}\bar{\boldsymbol{X}}^{(k)}\boldsymbol{B} - \boldsymbol{C}\bar{\boldsymbol{X}}^{(k)}\boldsymbol{D}, \quad k = 0, 1, \dots, t - 1. \tag{25}$$

Through approximation, computation overhead is greatly reduced. See details in the Appendix A.

As far as we are concerned, the advantages of utilizing feature variance regularization are three-fold. First, the regularizer measures the difference between features, therefore explicitly preventing all features from falling into the same subspace. Second, the modified convolution filter does not require additional training parameters, avoiding the risk of overfitting. Third, the regularizer is designed based on the proposed unified framework, which means it can be easily implemented on other convolution-based networks as a plug-in module.

## 4.2 DISCUSSION

Several researches have also shared insights on understanding and tackling oversmoothing. It is shown in (Li et al., 2018) that the graph convolution of GCN is a special form of Laplacian smoothing and the authors try to compensate the long-range dependencies by co-training GCN with a random walk model. JKNet (Xu et al., 2018b) proved that the influence score between nodes converges to a fixed distribution when layer stacks, therefore losing local information. As a remedy, they proposed to concatenate layer-wise representations to perform mixed structural information. More recently, Oono & Suzuki (2020) theoretically demonstrated that graph neural networks lose expressive power exponentially due to oversmoothing. Comparing to the aforementioned researches, our proposed method acts explicitly on the graph signals and can be easily implemented on other convolution-based networks as a plug-in module with trivial adaptations.

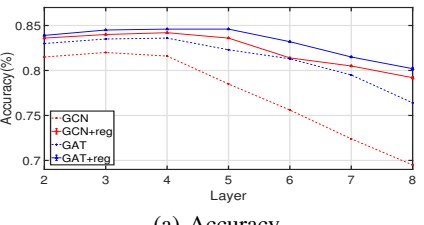 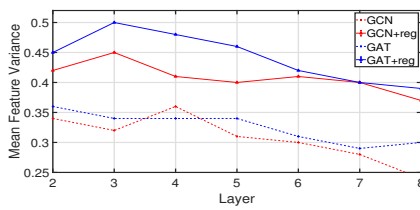

(a) Accuracy  (b) Mean Feature Variance

Figure 1: Accuracy and mean feature variance on Cora. Use GCN and GAT for comparison.

## 4.3 EXPERIMENT

To testify the effectiveness of the regularizer, we empirically validate the proposed method on several widely used semi-supervised node classification benchmarks, including transductive and inductive settings. As we have stated in Section 4.1, our regularizer can be implemented on various convolution-based approaches under the unified graph convolution framework. Therefore, we consider three different versions by implementing the regularizer on vanilla-GCNs, attention-based GCNs and topology-based GCNs. We achieve state-of-the-art results on almost all of the settings and show the effectiveness of tackling oversmoothing on graph-structured data.

**Dataset and Experimental Setup.** We conduct experiments on four real-world graph datasets. For transductive learning, we evaluate our method on the Cora, Citeseer, Pubmed datasets, following the experimental setup in (Sen et al., 2008). PPI (Zitnik & Leskovec, 2017) is adopted for inductive learning. Dataset statistics and more experimental setups are presented in Appendix C. For comparison, we categorize state-of-the-art convolution-based graph neural networks into three specific classes, corresponding to the three versions of our proposed method. The first category is based on the vanilla-GCN proposed by Kipf & Welling (2017), including GCN, FastGCN (Chen et al., 2018), SGC (Wu et al., 2019a), GIN (Xu et al., 2018a), and DGI (Veličković et al., 2019). Since GIN is not initially evaluated on citation networks, we implement GIN following the setting in (Xu et al., 2018a). The second category corresponds to the attention-based approaches, including GAT (Veličković et al., 2018), AGNN (Thekumparampil et al., 2018), MoNet (Monti et al., 2017) and GatedGCN (Bresson & Laurent, 2017). The last category of approaches is topology-based GCNs which utilizes the structural information in the multi-hop neighborhood. We consider APPNP (Klicpera et al., 2018), TAGCN (Du et al., 2017) and MixHop (Kapoor et al., 2019) as the baselines.

**Transductive Learning.** Table 1 presents the performance of our method and several state-of-the-art graph neural networks on transductive learning datasets. For three classes of convolution-based approaches, we implement our regularizer with GCN, GAT and APPNP as comparisons with other baselines, respectively. For a fair comparison, we adopt the same network structure, hyperparameters and training configurations as baseline models. It is shown that the proposed model achieves state-of-the-art results on all three settings. On all of the datasets, we can observe a 0.5~1.0% higher performance after adopting the proposed reg-

Table 2: Test Micro-F1 Score on inductive learning dataset. We report mean values and standard deviations in 5 independent experiments.

| Dataset | PPI |
|---|---|
| GCN (Kipf & Welling, 2017) | 92.4 |
| GAT (Veličković et al., 2018) | 97.3 |
| SGC (Wu et al., 2019a) | 66.4 |
| JKNet (Xu et al., 2018b) | 97.6 |
| GraphSAGE (Hamilton et al., 2017) | 61.2 |
| DGI (Veličković et al., 2019) | 63.8 |
| **GCN+reg (ours)** | 97.69±0.32 |
| **GAT+reg (ours)** | **98.23±0.08** |

ularizer. Notably, the proposed model achieves the highest improvement on the vanilla GCNs as this simplest version suffers most from the oversmoothing problem. Meanwhile, when combining with GAT, the model achieves the highest results comparing with almost all the baselines. Considering that attention mechanism and the regularization on oversmoothing focus on the local and global properties respectively, this can be an ideal combination for graph representation learning. We also conduct experiments on three citation networks with random splits and present the result in Appendix D.

**Inductive Learning.** For the inductive learning task, we implement our method on the vanilla GCN and GAT, and adopt the same experimental setup. Table 2 presents the comparison results on

Table 3: Comparison results on transductive learning datasets. We report mean values and standard deviations in 30 independent experiments. The best results are highlighted with **boldface**. The number in the brackets represent the number of GCN layers when achieving the best performance.

| Method | Citeseer | Cora | Pubmed |
|---|---|---|---|
| GCN + DropEdgeRong et al. (2019) | 73.2±0.1 (4) | 84.2±0.6 (6) | 80.1±0.3 (6) |
| GCN + PairNormZhao & Akoglu (2019) | 71.0±0.5 (3) | 82.2±0.6 (2) | 79.6±0.6 (4) |
| **GCN + regs (ours)** | **73.6±0.4 (4)** | **84.9±0.2 (5)** | **81.0±0.4 (5)** |

inductive learning dataset. It can be seen that our model compares favorably with all the competitive baselines. On the PPI dataset, out model achieves 0.5∼1% higher on test Micro-F1 score, showing the effectiveness of applying our method under inductive settings.

**Comparison with Other Related Works.** To validate the effectiveness of our model, we compare the proposed regularizer with two state-of-the-art approaches on tackling oversmoothing, DropEdge(Rong et al., 2019) and PairNorm(Zhao & Akoglu, 2019). For fair comparison, all approaches are adopted on vanilla-GCN with 2∼8 layers and show the best performance on three transductive datasets respectively. As shown in Table 3, our regularizer achieves best performance on all three settings. As PairNorm is more suitable when a subset of the nodes lack feature vectors, it is less competitive in the general settings.

**Analysis.** As we have stated above, the regularizer can be interpreted as the mean feature variance, which prevents different features from falling into the same subspace. To testify the effect of our method, we compute the mean pairwise distance (Eq.(19)) of the last hidden layer of GCN and GAT, with and without regularizer on the Cora dataset. We show the result of models with 2-8 layers in Figure 1. As we can observe, the feature variances and the accuracies of models with regularization are comparably higher than vanilla models with obvious gaps. Therefore, after applying the regularizer, features are more separated from each other, and the oversmoothing problem is alleviated.

## 5 CONCLUSION

In this paper, we develop a unified graph convolution framework by establishing graph convolution filters with optimization problems in the graph Fourier space. We show that most convolution-based graph learning models are equivalent to adding carefully designed regularizers. Besides vanilla GCN, our framework is extended to formulating non-convolutional operations, attention-based GCNs and topology-based GCNs, which cover a large fraction of state-of-the-art graph learning models. On this basis, we propose a novel regularization on tackling the oversmoothing problem as a showcase, proving the effectiveness of designing new modules based on the framework. Through the unified framework, we provide a general methodology for understanding and relating different graph learning modules, with new insights on tackling common problems and improving the generalization performance of current graph neural networks in the graph Fourier domain. Meanwhile, the unified framework can also serve as a once-for-all platform for expert-designed modules on convolution-based approaches. We hope our work can promote the understandings towards graph convolutional networks and inspire more insights in this field.

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

APPENDIX

## A. PROOFS OF THE DEFINITIONS

**Definition 2 Vanilla GCNs.** *Let* $\bar{\boldsymbol{x}}(i)_{i\in\mathbb{V}}$ *be the estimation of the input observation* $\boldsymbol{x}(i)_{i\in\mathbb{V}}$. *A low-pass filter:*

$$\bar{\boldsymbol{X}} = \tilde{\mathcal{A}}_{\mathrm{rw}}\boldsymbol{X}, \tag{26}$$

*is the first-order approximation of the optimal solution of the following optimization:*

$$\min_{\bar{\boldsymbol{X}}} \sum_{i\in\mathbb{V}} \|\bar{\boldsymbol{x}}(i) - \boldsymbol{x}(i)\|^2_{\tilde{\boldsymbol{D}}} + \sum_{i,j\in\mathbb{V}} a_{ij}\|\bar{\boldsymbol{x}}(i) - \bar{\boldsymbol{x}}(j)\|^2_2. \tag{27}$$

*Proof.* Let $l$ denote the objective function. We have

$$l = \mathrm{tr}[(\bar{\boldsymbol{X}} - \boldsymbol{X})^T \tilde{\boldsymbol{D}}(\bar{\boldsymbol{X}} - \boldsymbol{X})] + \mathrm{tr}(\bar{\boldsymbol{X}}^T \boldsymbol{L}\bar{\boldsymbol{X}}).$$

Then,

$$\frac{\partial l}{\partial \bar{\boldsymbol{X}}} = 2\tilde{\boldsymbol{D}}(\bar{\boldsymbol{X}} - \boldsymbol{X}) + 2\boldsymbol{L}\bar{\boldsymbol{X}}.$$

If we let $\frac{\partial l}{\partial \bar{\boldsymbol{X}}} = 0$:

$$(\tilde{\boldsymbol{D}} + \boldsymbol{L})\bar{\boldsymbol{X}} = \tilde{\boldsymbol{D}}\boldsymbol{X}$$

$$(\boldsymbol{I} + \tilde{\mathcal{L}}_{\mathrm{rw}})\bar{\boldsymbol{X}} = \boldsymbol{X}.$$

As the norm of eigenvalues of $\tilde{\mathcal{A}}_{\mathrm{rw}} = \boldsymbol{I} - \tilde{\mathcal{L}}_{\mathrm{rw}}$ is bounded by 1, $\boldsymbol{I} + \tilde{\mathcal{L}}_{\mathrm{rw}}$ has eigenvalues in range [1, 3], which proves that $\boldsymbol{I} + \tilde{\mathcal{L}}_{\mathrm{rw}}$ is a positive definite matrix. Therefore,

$$\bar{\boldsymbol{X}} = (\boldsymbol{I} + \tilde{\mathcal{L}}_{\mathrm{rw}})^{-1}\boldsymbol{X}. \tag{28}$$

Unfortunately, solving the closed-form solution of Eq.(28) is computationally expensive. Nevertheless, we can derive a simpler form, $\bar{\boldsymbol{X}} \approx (\boldsymbol{I} - \tilde{\mathcal{L}}_{\mathrm{rw}})\boldsymbol{X} = \tilde{\mathcal{A}}_{\mathrm{rw}}\boldsymbol{X}$, via first-order Taylor approximation which establishes the Definition. □

**Definition 3 Residual Connection.** *A graph convolution filter with residual connection:*

$$\bar{\boldsymbol{X}} = \tilde{\mathcal{A}}_{\mathrm{rw}}\boldsymbol{X} + \epsilon\boldsymbol{X}, \tag{29}$$

*where* $\epsilon > 0$ *controls the strength of residual connection, **is the first-order approximation of** the optimal solution of the following optimization:*

$$\min_{\bar{\boldsymbol{X}}} \sum_{i\in\mathbb{V}}(\|\bar{\boldsymbol{x}}(i) - \boldsymbol{x}(i)\|^2_{\tilde{\boldsymbol{D}}} - \epsilon\|\bar{\boldsymbol{x}}(i)\|^2_{\tilde{\boldsymbol{D}}}) + \sum_{i,j\in\mathbb{V}} a_{ij}\|\bar{\boldsymbol{x}}(i) - \bar{\boldsymbol{x}}(j)\|^2_2. \tag{30}$$

*Proof.* Let $l$ denote the objective function. We have

$$l = \mathrm{tr}[(\bar{\boldsymbol{X}} - \boldsymbol{X})^T \tilde{\boldsymbol{D}}(\bar{\boldsymbol{X}} - \boldsymbol{X})] - \epsilon\mathrm{tr}(\bar{\boldsymbol{X}}^T \tilde{\boldsymbol{D}}\bar{\boldsymbol{X}}) + \mathrm{tr}(\bar{\boldsymbol{X}}^T \boldsymbol{L}\bar{\boldsymbol{X}}).$$

Then,

$$\frac{\partial l}{\partial \bar{\boldsymbol{X}}} = 2\tilde{\boldsymbol{D}}(\bar{\boldsymbol{X}} - \boldsymbol{X}) + 2(\boldsymbol{L} - \epsilon\tilde{\boldsymbol{D}})\bar{\boldsymbol{X}}.$$

If we let $\frac{\partial l}{\partial \bar{\boldsymbol{X}}} = 0$:

$$[(1 - \epsilon)\tilde{\boldsymbol{D}} + \boldsymbol{L}]\bar{\boldsymbol{X}} = \tilde{\boldsymbol{D}}\boldsymbol{X}$$

$$\bar{\boldsymbol{X}} = [(1 - \epsilon)\boldsymbol{I} + \tilde{\mathcal{L}}_{\mathrm{rw}}]^{-1}\boldsymbol{X}$$

$$\bar{\boldsymbol{X}} = [\boldsymbol{I} + (\tilde{\mathcal{L}}_{\mathrm{rw}} - \epsilon\boldsymbol{I})]^{-1}\boldsymbol{X}.$$

Therefore, the first-order approximation of the optimal solution is

$$\bar{\boldsymbol{X}} \approx [\boldsymbol{I} - (\tilde{\mathcal{L}}_{\mathrm{rw}} - \epsilon\boldsymbol{I})]\boldsymbol{X}$$

$$= \tilde{\mathcal{A}}_{\mathrm{rw}}\boldsymbol{X} + \epsilon\boldsymbol{X}.$$

□

**Definition 3' Concatenation.** *A graph convolution filter concatenating with the input signal:*

$$\bar{X} = \tilde{\mathcal{A}}_{\mathrm{rw}} X + \epsilon X \Theta \Theta^T, \tag{31}$$

*is the first-order approximation of the optimal solution of the following optimization:*

$$\min_{\bar{X}} \ \sum_{i \in \mathbb{V}} (\|\bar{x}(i) - x(i)\|_{\tilde{D}}^2 - \epsilon \|\bar{x}(i)\Theta\|_{\tilde{D}}^2) + \sum_{i,j \in \mathbb{V}} a_{ij} \|\bar{x}(i) - \bar{x}(j)\|_2^2, \tag{32}$$

*where $\epsilon > 0$ controls the strength of concatenation and $\Theta$ is the learning coefficients for the concatenated signal.*

*Proof.* Let $l$ denote the objective function. We have

$$l = \mathrm{tr}[(\bar{X} - X)^T \tilde{D} (\bar{X} - X)] - \epsilon \mathrm{tr}((\bar{X}\Theta)^T \tilde{D}(\bar{X}\Theta)) + \mathrm{tr}(\bar{X}^T L \bar{X}).$$

Then,

$$\frac{\partial l}{\partial \bar{X}} = 2\tilde{D}(\bar{X} - X) + 2L\bar{X} - 2\epsilon \tilde{D}\bar{X}\Theta\Theta^T.$$

If we let $\frac{\partial l}{\partial \bar{X}} = 0$:

$$(\tilde{D} + L)\bar{X} - \epsilon \tilde{D}\bar{X}\Theta\Theta^T = \tilde{D}X$$
$$(I + \tilde{\mathcal{L}}_{\mathrm{rw}})\bar{X} - \epsilon \bar{X}\Theta\Theta^T = X.$$

With the help of the Kronecker product operator $\otimes$ and first-order Taylor expansion, we have

$$\begin{aligned}
\mathrm{vec}(\bar{X}) &= [(I \otimes (I + \tilde{\mathcal{L}}_{\mathrm{rw}})) - \epsilon((\Theta\Theta^T) \otimes I)]^{-1}\mathrm{vec}(X) \\
&\approx [2I - (I \otimes (I + \tilde{\mathcal{L}}_{\mathrm{rw}})) + \epsilon((\Theta\Theta^T) \otimes I)]\mathrm{vec}(X) \\
&= \mathrm{vec}(2X - (I + \tilde{\mathcal{L}}_{\mathrm{rw}})X + \epsilon\bar{X}\Theta\Theta^T) \\
&= \mathrm{vec}(\tilde{A}_{rw}X + \epsilon X\Theta\Theta^T).
\end{aligned}$$

$\square$

**Definition 4 Attention-based GCNs.** *An attention-based graph convolution filter:*

$$\bar{X} = PX, \tag{33}$$

*is the first-order approximation of the optimal solution of the following optimization:*

$$\min_{\bar{X}} \ \sum_{i \in \mathbb{V}} \|\bar{x}(i) - x(i)\|_{\tilde{D}}^2 + \sum_{i,j \in \mathbb{V}} p_{ij}\|\bar{x}(i) - \bar{x}(j)\|_2^2, \qquad s.t. \sum_{j \in \mathbb{V}} p_{ij} = \tilde{D}_{ii}, \forall i \in \mathbb{V}. \tag{34}$$

*Proof.* Let $l$ denote the objective function. We have

$$l = \mathrm{tr}[(\bar{X} - X)^T \tilde{D}(\bar{X} - X)] + \mathrm{tr}(\bar{X}^T L \bar{X}).$$

Then,

$$\frac{\partial l}{\partial \bar{X}} = 2\tilde{D}(\bar{X} - X) + 2(\tilde{D} - \tilde{D}P)\bar{X}.$$

If we let $\frac{\partial l}{\partial \bar{X}} = 0$:

$$(2\tilde{D} - \tilde{D}P)\bar{X} = \tilde{D}X$$
$$(2I - P)\bar{X} = X.$$

Similarly, we can prove that $(2I - P)$ is a positive definite matrix, with eigenvalues in range $[1, 3]$. Therefore,

$$\bar{X} = (2I - P)^{-1}X$$
$$\approx PX.$$

$\square$

**Definition 5 & 6 Topology-based GCNs** Due to the fact that most of the topology-based models adopt non-convolutional operations like concatenation, we derive a more general objective function by combining with the non-convolutional operations:

$$\min_{\bar{\boldsymbol{X}}} \quad \alpha_0 \sum_{i \in \mathbb{V}} \|\bar{\boldsymbol{x}}(i) - \boldsymbol{x}(i)\|_{\tilde{\boldsymbol{D}}}^2 + \sum_{k=1}^{t} \alpha_k \sum_{i,j \in \mathbb{V}} a_{ij}^{(k)} \|\bar{\boldsymbol{x}}(i)\Theta^{(k)} - \bar{\boldsymbol{x}}(j)\Theta^{(k)}\|_2^2, \tag{35}$$

where $\sum_{k=0}^{t} \alpha_k = 1$, $\alpha_0 > 0$ and $\alpha_k \geq 0, k = 1, 2, \ldots, t$. If we let $d$ be the feature dimension of $\boldsymbol{X}$, $\Theta^{(k)} \in \mathbb{R}^{d \times d}$ correspond to the learning weights for the $k_{th}$ hop neighborhood. Let $l$ denote the objective function, we have:

$$\frac{\partial l}{\partial \bar{\boldsymbol{X}}} = \alpha_0 \tilde{\boldsymbol{D}}(\bar{\boldsymbol{X}} - \boldsymbol{X}) + \sum_{k=1}^{t} \alpha_k (\tilde{\boldsymbol{D}} - \tilde{\boldsymbol{D}}\tilde{\mathcal{A}}_{rw}^k)\bar{\boldsymbol{X}}\Theta^{(k)}(\Theta^{(k)})^T.$$

By letting $\frac{\partial l}{\partial \bar{\boldsymbol{X}}} = 0$, we have:

$$\alpha_0 \bar{\boldsymbol{X}} + \sum_{k=1}^{t} (\boldsymbol{I}_n - \tilde{\mathcal{A}}_{rw}^k)\bar{\boldsymbol{X}}\Theta^{(k)}(\Theta^{(k)})^T = \alpha_0 \boldsymbol{X}.$$

Therefore, with the help of the Kronecker product operator $\otimes$ and first-order Taylor expansion, we have

$$[\alpha_0 \boldsymbol{I}_n + \sum_{k=1}^{t} (\alpha_k \Theta^{(k)}(\Theta^{(k)})^T) \otimes (\boldsymbol{I}_n - \tilde{\mathcal{A}}_{rw}^k)]\mathrm{vec}(\bar{\boldsymbol{X}}) = \alpha_0 \mathrm{vec}(\boldsymbol{X}). \tag{36}$$

We can observe that $\sum_{k=1}^{t} (\alpha_k \Theta^{(k)}(\Theta^{(k)})^T)$ and $(\boldsymbol{I}_n - \tilde{\mathcal{A}}_{rw}^k)$ have non-negative eigenvalues. Due to the property of the Kronecker product that the eigenvalues of the Kronecker product $(\boldsymbol{A} \otimes \boldsymbol{B})$ equal to the product of eigenvalues of $\boldsymbol{A}$ and $\boldsymbol{B}$, the filter $(\alpha_0 \boldsymbol{I}_n + \sum_{k=1}^{t} (\alpha_k \Theta^{(k)}(\Theta^{(k)})^T)$ is proved to be a positive definite matrix. Therefore,

$$\mathrm{vec}(\bar{\boldsymbol{X}}) = \alpha_0 [\alpha_0 \boldsymbol{I}_n + \sum_{k=1}^{t} (\alpha_k \Theta^{(k)}(\Theta^{(k)})^T) \otimes (\boldsymbol{I}_n - \tilde{\mathcal{A}}_{rw}^k)]^{-1} \mathrm{vec}(\boldsymbol{X})$$

$$\approx \alpha_0 [(2 - \alpha_0)\boldsymbol{I}_n - \sum_{k=1}^{t} (\alpha_k \Theta^{(k)}(\Theta^{(k)})^T) \otimes (\boldsymbol{I}_n - \tilde{\mathcal{A}}_{rw}^k)]\mathrm{vec}(\boldsymbol{X})$$

$$= \alpha_0 \mathrm{vec}[(2 - \alpha_0)\boldsymbol{X} - \sum_{k=1}^{t} \alpha_k (\boldsymbol{I}_n - \tilde{\mathcal{A}}_{rw}^k)\boldsymbol{X}\Theta^{(k)}(\Theta^{(k)})^T].$$

If we let

$$\boldsymbol{W}^{(0)} = \frac{2 - \alpha_0}{\alpha_0}\boldsymbol{I}_n - \sum_{k=1}^{t} \frac{\alpha_k}{\alpha_0}\Theta^{(k)}(\Theta^{(k)})^T; \tag{37}$$

$$\boldsymbol{W}^{(k)} = \Theta^{(k)}(\Theta^{(k)})^T), \quad k = 1, 2, \ldots, t; \tag{38}$$

we can denote the convolution filter as:

$$\bar{\boldsymbol{X}} = \sum_{k=0}^{t} \alpha_k \tilde{\mathcal{A}}_{rw}^k \boldsymbol{X}\boldsymbol{W}^{(k)}. \tag{39}$$

As we have stated in the Section 2.2.2, although the learning weights has a constrained expressive capability, it can be compensated by the following feature learning module. We omit the proofs of Definition 5 and 6, as they can be viewed as particular instances of (35).

**Definition 7 Regularized Feature Variance.** *Let $\otimes$ be the Kronecker product operator, $\mathrm{vec}(\boldsymbol{X}) \in \mathbb{R}^{nd}$ be the vectorized signal $\boldsymbol{X}$. Let $\boldsymbol{D}_X$ be a diagonal matrix whose value is defined by $\boldsymbol{D}_X(i,i) = \|x_{\cdot i}\|_2$. A graph convolution filter with regularized feature variance:*

$$\mathrm{vec}(\bar{\boldsymbol{X}}) = (\boldsymbol{I}_n \otimes [(\alpha_1 + \alpha_2)I - \alpha_2 \tilde{\mathcal{A}}_{rw}] - \alpha_3 [\boldsymbol{D}_x^{-1}(\boldsymbol{I} - \frac{1}{d}\boldsymbol{1}\boldsymbol{1}^T)\boldsymbol{D}_x^{-1}] \otimes \tilde{\boldsymbol{D}}^{-1})^{-1}\mathrm{vec}(\boldsymbol{X}) \tag{40}$$

*is equivalent to* the optimal solution of the following optimization:

$$\min_{\bar{X}} \alpha_1 \sum_{i \in \mathbb{V}} \|\bar{x}(i) - x(i)\|_{\bar{D}}^2 + \alpha_2 \sum_{i,j \in \mathbb{V}} a_{ij} \|\bar{x}(i) - \bar{x}(j)\|_2^2 - \alpha_3 \frac{1}{d} \sum_{i,j \in d} \|\bar{x}_{\cdot i}/\|x_{\cdot i}\| - \bar{x}_{\cdot j}/\|x_{\cdot j}\|\|_2^2,$$

(41)

where $\alpha_1 > 0$, $\alpha_2, \alpha_3 \geq 0$. For computation efficiency, we approximate $D_{\bar{X}}$ with $D_X$ as we assume that a single convolution filter provides little effect to the norm of features.

*Proof.* Let $l$ denote the objective function. We have

$$l = \alpha_1 \mathrm{tr}[(\bar{X} - X)^T \tilde{D}(\bar{X} - X)] + \alpha_2 (\bar{X}^T L \bar{X}) - \alpha_3 \mathrm{tr}[\bar{X} D_x^{-1}(I - \frac{1}{d}\mathbf{1}\mathbf{1}^T)D_x^{-1}\bar{X}^T].$$

Then,

$$\frac{\partial l}{\partial \bar{X}} = 2\alpha_1 \tilde{D}(\bar{X} - X) + 2\alpha_2 L\bar{X} - 2\alpha_3 \bar{X} D_x^{-1}(I - \frac{1}{d}\mathbf{1}\mathbf{1}^T)D_x^{-1}.$$

If we let $\frac{\partial l}{\partial \bar{X}} = 0$:

$$[(\alpha_1 + \alpha_2)I - \alpha_2 \tilde{D}^{-1}\tilde{\mathcal{A}}_{\mathrm{rw}}]\bar{X} - \alpha_3 \tilde{D}^{-1}\bar{X}D_x^{-1}(I - \frac{1}{d}\mathbf{1}\mathbf{1}^T)D_x^{-1} = \alpha_1 X.$$

With the help of the Kronecker product operator $\otimes$, we have

$$(I_n \otimes [(\alpha_1 + \alpha_2)I - \alpha_2 \tilde{\mathcal{A}}_{\mathrm{rw}}] - \alpha_3[D_x^{-1}(I - \frac{1}{d}\mathbf{1}\mathbf{1}^T)D_x^{-1}] \otimes \tilde{D}^{-1})\mathrm{vec}(\bar{X}) = \mathrm{vec}(X). \quad (42)$$

By setting $\alpha_3$ with a small positive value, the filter in Eq.(42) is still a positive definite matrix. Therefore we complete the proof. $\qquad\square$

Similarly, we can derive a simpler form via Taylor approximation. If we let:

$$A = (\alpha_1 + \alpha_2)I - \alpha_2 \tilde{\mathcal{A}}_{\mathrm{rw}}, \qquad B = I_n, \tag{43}$$

$$C = -\alpha_3 \tilde{D}^{-1}, \qquad D = D_x^{-1}(1 - \frac{1}{d}\mathbf{1}\mathbf{1}^T)D_x^{-1}. \tag{44}$$

Then, the first-order approximation of Eq.(40) is summarized as:

$$\begin{aligned}
\mathrm{vec}(\bar{X}) &= (B^T \otimes A + D^T \otimes C)^{-1}\mathrm{vec}(X) \\
&\approx (2I - B^T \otimes A - D^T \otimes C)\mathrm{vec}(X) \\
&= \mathrm{vec}(2X - AXB - CXD).
\end{aligned}$$

Additionally, we can also derive a t-order approximated formulation:

$$\mathrm{vec}(\bar{X}^{(t)}) = (I + \sum_{i=1}^{t}[I - (B^T \otimes A + D^T \otimes C)]^i)\mathrm{vec}(X).$$

However, it is computationally expensive to calculate the Kronecker product. Therefore, we consider utilizing a iterative algorithm. For any $0 \leq k < t$

$$\begin{aligned}
\mathrm{vec}(\bar{X}^{(k+1)}) &= (I + \sum_{i=1}^{k+1}[I - (B^T \otimes A + D^T \otimes C)]^i)\mathrm{vec}(X) \\
&= [I - (B^T \otimes A + D^T \otimes C)](I + \sum_{i=1}^{k}[I - (B^T \otimes A + D^T \otimes C)]^i)\mathrm{vec}(X) + \mathrm{vec}(X) \\
&= [I - (B^T \otimes A + D^T \otimes C)]\mathrm{vec}(\bar{X}^{(k)}) + \mathrm{vec}(X) \\
&= \mathrm{vec}(X + \bar{X}^{(k)} - A\bar{X}^{(k)}B - C\bar{X}^{(k)}D).
\end{aligned} \tag{45}$$

## B. REFORMULATION EXAMPLES

The reformulation examples of GCN derivatives are presented in Table 4.

Table 4: Reformulation of convolution-based graph neural networks. $D$ and $d_i$ in the attention-based modules are normalization coefficients.

| Models | Non-Conv Module | Attention-based Module | Topology-based Module |
|---|---|---|---|
| GIN (Xu et al., 2018a) | Residual Connection | - | - |
| GraphSAGE (Hamilton et al., 2017) | Concatenation | - | - |
| RGCN (Schlichtkrull et al., 2018) | Concatenation $W = \sum_{r \in \mathbb{R}} \frac{1}{c_r} W_r$ | - | - |
| SplineCNN (Fey et al., 2018) | - | $p_{ij} = h_\theta(e_{ij})$ | - |
| ECC (Simonovsky & Komodakis, 2017) | Concatenation | $p_{ij} = h_\theta(e_{ij})$ | - |
| AGNN (Thekumparampil et al., 2018) | - | $p_{ij} = d_i \frac{\exp(\beta cos(\boldsymbol{x}_i, \boldsymbol{x}_j))}{\sum_{k \in \mathbb{N}(i) \cup i} \exp(\beta cos(\boldsymbol{x}_i, \boldsymbol{x}_k))}$ | - |
| MoNet (Monti et al., 2017) | Concatenation | $p_{ij}^{(k)} = d_i \exp(-\frac{1}{2}(e_{ij} - \mu_k)^T \Sigma_k^{(-1)}(e_{ij} - \mu_k))$ | - |
| GAT Veličković et al. (2018) | Concatenation | $p_{ij}^{(k)} = d_i \frac{\exp(\sigma(a_{(k)}^T [\theta \boldsymbol{x}_i \| \theta \boldsymbol{x}_j]))}{\sum_{k \in \mathbb{N}(i) \cup i} \exp(\sigma(a_{(k)}^T [\theta \boldsymbol{x}_i \| \theta \boldsymbol{x}_k]))}$ | - |
| Cluster GCN (Chiang et al., 2019) | Concatenation | $P = D(\tilde{A}_{\mathrm{rw}} + \lambda \mathrm{diag}(\tilde{A}_{\mathrm{rw}}))$ | |
| SGC (Wu et al., 2019a) | - | $P = D\tilde{A}_{\mathrm{sym}}^k$ | - |
| Hyper-Atten (Bai et al., 2019) | - | $P = HWB^{-1}H^T$ | - |
| APPNP (Klicpera et al., 2018) | - | - | $\alpha_0 = \gamma, \alpha_1 = 1 - \gamma$ |
| GDC (Klicpera et al., 2019) | - | - | $\alpha_i = \theta_i$ |
| TAGCN (Du et al., 2017) | - | - | $\alpha_0 = \cdots = \alpha_k = 1/(k+1)$ |
| MixHop (Kapoor et al., 2019) | Concatenation | - | $\alpha_0 = \alpha_1 = \alpha_2 = 1/3$ |

Table 5: Dataset Statistics

| Dataset | Cora | Citeseer | Pubmed | PPI |
|---|---|---|---|---|
| Nodes | 2,708 | 3,327 | 19,717 | 56,944(24 graphs) |
| Edges | 5,429 | 4,732 | 44,338 | 818,716 |
| Features | 1,433 | 3,703 | 500 | 50 |
| Classes | 7 | 6 | 3 | 121(multilabel) |
| Training Nodes | 140 | 120 | 60 | 44,906(20 graphs) |
| Validation Nodes | 500 | 500 | 500 | 6,514(2 graphs) |
| Test Nodes | 1,000 | 1,000 | 1,000 | 5,524(2 graphs) |

## C. DATA STATISTICS AND EXPERIMENTAL SETUPS

We conduct experiments on four real-world graph datasets, whose statistics are listed in Table 5. For transductive learning, we evaluate our method on the Cora, Citeseer, Pubmed datasets, following the experimental setup in (Sen et al., 2008). There are 20 nodes per class with labels to be used for training and all the nodes' features are available. 500 nodes are used for validation and the generalization performance is tested on 1000 nodes with unseen labels. PPI (Zitnik & Leskovec, 2017) is adopted for inductive learning, which is a protein-protein interaction dataset containing 20 graphs for training, 2 for validation and 2 for testing while testing graphs remain unobserved during training.

To ensure a fair comparison with other methods, we implement our module without interfering the original network structure. In all three settings, we use two convolution layers with hidden dimen-

Table 6: Test accuracy (%) on transductive learning datasets with random slits. We report mean values and standard deviations of the test accuracies over 100 random train/validation/test splits.

| Dataset | Citeseer | Cora | Pubmed |
|---|---|---|---|
| GCN(Kipf & Welling, 2017) | 71.9±1.9 | 81.5±1.3 | 77.8±2.9 |
| GAT(Veličković et al., 2018) | 71.4±1.9 | 81.8±1.3 | 78.7±2.3 |
| MoNet (Monti et al., 2017) | 71.2±2.0 | 81.3±1.3 | 78.6±2.3 |
| GraphSAGE (Hamilton et al., 2017) | 71.6±1.9 | 79.2±7.7 | 77.4±2.2 |
| **GCN+reg (ours)** | **72.9±1.4** | **83.6±1.2** | **79.9±1.6** |

Table 7: Training and test time on Cora. We report mean values in 5 independent experiments. The best results are highlighted with **boldface**.

| Method | Training Time (s) | Training Time (ms / epoch) | Test Time (ms) |
|---|---|---|---|
| GCN (Kipf & Welling, 2017) | **1.8** | **3.8** | **1.9** |
| GAT (Veličković et al., 2018) | 5.4 | 8.5 | 3.3 |
| AGNN (Thekumparampil et al., 2018) | 5.3 | 7.9 | 3.2 |
| APPNP (Klicpera et al., 2018) | 9.8 | 14.2 | 13.6 |
| GCN + regs (ours) | 5.7 | 8.0 | 3.2 |

sion $h = 64$. We set $\alpha_1 = 0.2$, $\alpha_2 = 0.8$ and $\alpha_3 = 0.05$ for all four datasets. We apply $L_2$ regularization with $\lambda = 0.0005$ and use dropout on both layers. For training strategy, we initialize weights using the initialization described in (Glorot & Bengio, 2010) and follow the method proposed in GCN, adopting an early stop if validation loss does not decrease for certain consecutive epochs. The implementations of baseline models are based on the PyTorch-Geometric library (Fey & Lenssen, 2019) in all experiments.

## D. RANDOM SPLITS

As illustrated in (Shchur et al., 2018), using the same train/validation/test splits of the same datasets precludes a fair comparison of different architectures. Therefore, we follow the setup in (Shchur et al., 2018) and evaluate the performance of our model on three citation networks with random splits. Empirically, for each dataset, we use 20 labeled nodes per class as the training set, 30 nodes per class as the validation set, and the rest as the test set. For every model, we choose the hyperparameters that achieve the best average accuracy on Cora and CiteSeer datasets and applied to Pubmed dataset.

Table 6 shows the results on three citation networks under the random split setting. As we can observe, our model consistently achieves higher performances on all the datasets. On Citeseer, our model achieves higher accuracy than on the original split. On Cora and Pubmed, the test accuracies of our model are comparable to the original split, while most of the baselines suffer from a serious decline.

## E. TIME CONSUMPTION

As we have shown in Eq.(20), the computation of graph filter with the regularizer is greatly increased with Kronecker product and inverse matrix operations. Nevertheless, we approximate the filter with an iterative algorithm as stated in Eq.(25) and realize an efficient implement. To empirically testify the computation efficiency, we conduct experiments on Cora and report the training and test time of several GCN models on a single RTX 2080 Ti GPU. Due to the early stopping rule (see details in Appendix C), the training epoch for each module is different. The results are shown in Table 7. As we can observe, when combining with vanilla GCNs, the training and test time of our model is similar to GAT and AGNN and faster than APPNP.

Table 8: Ablation study on the regularization strength. We report mean values and standard deviations in 30 independent experiments. The best results are highlighted with **boldface**.

| Regularization Strength | Citeseer | Cora | Pubmed |
|---|---|---|---|
| 0 | 70.3±0.4 | 81.5±0.5 | 79.0±0.4 |
| 0.01 | 71.7±0.6 | 83.0±0.5 | 79.2±0.6 |
| 0.05 | 72.2±0.4 | **83.6±0.3** | **79.8±0.2** |
| 0.1 | **72.4±0.5** | 83.5±0.2 | 79.4±0.7 |
| 0.2 | 68.4±1.0 | 76.8±1.3 | 70.2±2.2 |

## F. ABLATION STUDY

To analyze the effects of the regularization strength, we conduct experiments on three transductive datasets and present the results in Table 8. As we can observe, with reasonable choice of the regularization strength, our approach can achieve consistent improvement under all settings. However, when the regularization strength is too large, the training procedure becomes unstable and the model performance suffers from a severe decrease.

