# OpenReview forum: "A Unified Framework for Convolution-based Graph Neural Networks"
_ICLR.cc/2021/Conference — Reject_

### Official Review · AnonReviewer1 · 2020-10-21
**Good work on establishing the relationship between different GNN models**

**Rating:** 7
**Confidence:** 3

**Review:**

The paper introduces a unified framework for graph convolutional networks by interpreting filters as regularizers in the graph Fourier domain.
In particular, this framework allows to establish the relationships between standard, attention-based and topology-based GNNs.
Furthermore, the authors propose a regularization technique based upon the proposed framework which tackles the oversmoothing problem of GNNs, which achieves clear benefits on standard (small) benchmark datasets.

The paper is mostly well-written, although though to understand on first read.
I especially liked that it tries to establish a systematic view of different GNN models and their relations, which is a welcome work in the field of graph representation learning (especially with the sheer amount of GNN models available in literature).
In my opinion, the proposed framework has the potential to improve our understanding of GCNs and inspire better models in return.

On the other hand, it is not exactly clear to me how the proposed regularization technique differs from PairNorm (which is build upon similar insights by preventing node embeddings from becoming too similar). I would very much welcome a discussion between key differences and similarities between the two approaches. Furthermore, the authors should consider comparing the proposed regularization technique against related approaches, e.g., PairNorm and DropEdge.
Overall, the empirical evaluation feels a bit shallow by only evaluating on small benchmark datasets, but might be sufficient for a work that has mostly theoretical contributions.

Minor comments:

* It is not exactly clear to me how ECC can be viewed as an attention-based GNN since this operator learns a weight matrix conditioned on the edge features (instead of performing weighted normalization). Does this operator really fit into the proposed unified framework?

* A GCN baseline is missing on the PPI dataset.

============== Post Rebuttal Comments =================

I would like to thank the authors for their insightful rebuttal and clarifications. Sadly, I cannot find the newly added section regarding the non-linearity analysis in the revised manuscript and therefore cannot judge the findings of the authors.

Hence, my rating will stay the same.

---

> ### Author Response · Authors · 2020-11-21
> **Response to AnonReviewer1**
>
> We would first like to express our appreciation for your time and insightful comments. Please find our response to your concerns in the following:
>
> 1. **Connections between the proposed regularizer and PairNorm **
>
> The original paper of PairNorm introduces two metrics, which are referred to as **node-wise** and **feature-wise** oversmoothing (eq (3) & (4) in their paper), respectively. However, the PairNorm algorithm fully relies on the first metric, while our regularizer is inspired by the second. As these two metrics are essentially different, our approach differs from PairNorm substantially.
>
> Specifically, PairNorm is a **normalization technique** which designs a normalization layer to prevent unconnected **nodes** from getting closer. By comparison, ours is a **regularization technique** which introduces a regularizer to the objective function to maximize the distance between **features**. Moreover, PairNorm can be viewed as an individual layer, while our regularizer is incorporated in graph filters. Therefore, the two approaches tackle the oversmoothing problem from two different perspectives with different techniques, and can be viewed as complementary methods.
>
> In addition, we find that feature-wise distance is an effective metric for quantifying oversmoothing. As we have stated in section 4.1, features tend to fall into the same subspace spanned by the dominated eigenvectors of the normalized adjacency matrix. Similar observations are also found in [1][2], showing that the distance between features (rather than nodes) shrink to 0 with sufficient number of layers, which is the leading cause of oversmoothing.
>
> Empirically, PairNorm is more suitable when features are missing in the graph, where feature-wise oversmoothing is naturally alleviated. In comparison, our approach achieves consistent improvements in general settings when implemented on several SOTA GCNs.
>
> 2.	**Experiment section **
>
> We appreciate your suggestions and we have conducted experiments on comparisons with other related approaches. We update the additional results and Performance of GCN on PPI in the experiment (Section 4.3). We would also like to point out that our main contribution is the unified framework, and the proposed algorithm is for showing the feasibility of designing new regularizers based on the framework.
>
> 3.	** ECC formulation **
>
> Truly ECC is different from other attention-based approaches as the weight normalization is not employed. To address the unrestricted weight for each node, we can add a mapping matrix \Theta for signals in the attention regularizer as we have adopted in the concatenation formulation. (which turns the second form into \sum_{p_{ij}||x(i)\Theta-x(j)\Theta||^2}) With this adaptation, we can formulate unweighted attention matrices, e.g., ECC. We are sorry for the confusion and we add the reformulation in the Appendix B.
>
> Lastly, we are grateful for your time and we hope we have adequately addressed your concerns. We look forward to additional discussions.
>
> [1] Klicpera, Johannes et al. “Predict then Propagate: Graph Neural Networks meet Personalized PageRank.” ICLR (2019).
>
> [2] Li, Qimai et al. “Deeper Insights into Graph Convolutional Networks for Semi-Supervised Learning.” ArXiv abs/1801.07606 (2018): n. pag.

---

> > ### Comment · AnonReviewer1 · 2020-11-22
> > **Response to the authors**
> >
> > I would like to thank the authors for their rebuttal.
> >
> > I'm still of the opinion that this work is interesting and that it provides a good theoretical foundation for understanding GNNs by interpreting their various convolution filters as additional regularizers in the graph Fourier domain (while this might not be novel for vanilla-GCN, I am not aware of any work that extends these insights to other popular GNN variants):
> >
> > ## Pros:
> >
> > * Theoretical foundations to unify GCN-based models
> > * Proposal of a new regularization technique inspired by the introduced framework which seems to perform well in practice
> > * My concerns regarding the experimental evaluation have been addressed (Table 2 and Table 3) and look convincing to me (although the training split seems to be different from the experiments in Table 1).
> >
> > ## Cons:
> >
> > * As other reviewers reported, non-linearities are not considered which limits the impact of this work. As the authors state, this leads to substantial difficulty in theoretical analysis, but would strengthen the papers contribution and impact by a wide margin. In particular, since the importance of non-linearity in GCNs is still not well-understood as recent work indicates (e.g., SGC and APPNP).
> >
> > * Recent works on GNNs apply non-linear filters (e.g., ECC) or non-linear transformations (e.g., GIN). However, this work considers only linear operations or drops the non-linear transformation parts completely. While the proposed framework has the potential to unify linear filters, it does not look like it can do this for *any* graph-based convolutional filter (as the title claims).
> >
> > * I still don't see how ECC is compatible with the proposed framework (Eq. (18)), in particular because $\bar{x}(i)$ and $\bar{x}(j)$ should be transformed differently via $\Theta_{j,i}$ (see Eq. (1) in the ECC paper).

---

> > > ### Author Response · Authors · 2020-11-23
> > > **Further response to AnonReviewer1**
> > >
> > > Thanks for the comments. We appreciate your feedback and we’d like to take further discussions on the remaining concerns.
> > >
> > > ##Non-linearity ##
> > >
> > > To further address the concerns on the non-linearity issue, we update the paper with a newly added section (Sec. 4 in the latest version), which shows that under mild assumptions our main results are still applicable when non-linearity exists. Please refer to Sec.4 of the revised paper for detailed discussion.
> > >
> > > ##ECC formulation ##
> > >
> > > We update the formulation of ECC in the Appendix G. We find that categorizing ECC into the attention-based GCNs can lead to misunderstanding, Therefore, we remove the ECC from section 3.1.3 and only keep it in the Appendix B. Please let us know if you still have concerns.

---

### Official Review · AnonReviewer4 · 2020-10-27
**GNN unified using old ideas of regularisation, but non-linearity not accounted**

**Rating:** 5
**Confidence:** 3

**Review:**

Summary: The paper shows that several graph networks (GCN, attention GCN, PPNP, residual) can be unified under a common framework of Laplacian-regularised optimisation. Subsequently, different types of regularisation are combined to propose a new method for graph transduction, which is then empirically evaluated.

Significance: Laplacian regularisation is a classical approach for formulating/justifying graph transduction algorithms (multiple papers by Mikhail Belkin and Xiaojin Zhu around 2004-06). It is interesting to see that so many graph networks can also be unified in the same framework. A unified framework does aid in both theoretical analysis and implementation of GCNs.
However, the claims and derivation do not seem to account for the non-linear activation in the networks, and hence, significance of the work seems limited.

Quality: As noted above, non-linearity is not considered which makes the derivation significantly simpler. Moreover, the first-order approximation is quite misleading since even the proof do not seem to consider non-linear activation.
Since the proposed method combines multiple types of regularisation, it is expected to perform better than other networks. However, it is not clear if the training time increases due to the complex regularisation.

Clarity and orginality: The paper is otherwise well written / organised, and the theoretical contributions (although technically straightforward) seem original and somewhat interesting.

---

> ### Author Response · Authors · 2020-11-21
> **Response to AnonReviewer4**
>
> We would first like to express our appreciation for your time and insightful comments. Please find our response to your concerns in the following:
> 1.	** Non-linear activations **
>
> Thanks for the valuable suggestion. Nonlinear activation is indeed important for deep neural networks. Unfortunately, nonlinearity also causes substantial difficulty for theoretical analysis. Many existing works on analyzing GCNs [1][2] only explore the properties of linear operations but still provide thoughtful insights. In our paper we follow these researches and start with linear operations to understand and relate different GCN models.
> Despite the lack of nonlinear activations, we believe that our framework is still valuable for the following reasons:
>
> First, the key ingredient that differentiates many popular graph neural networks (like GCN, PPNP, and GDC) is the way of designing linear filters, while their nonlinear activation layers are identical (e.g., relu). To analyze the relations and differences among these approaches, it might be sufficient to consider only the linear operations. By formulating various GCNs from the optimization perspective, insights towards understanding and comparing the key components of different GCNs can still be provided.
>
> Second, recent work (e.g., SGC) has shown that even without nonlinear transformations, GCN can still yield competitive results.
>
> 2.	** Significance **
>
> Although we haven’t incorporated non-linear activations in the unified framework, we consider our work still makes valuable contributions from two perspectives. First, as we have stated in the ** Non-linear activations ** part, our framework can provide new insights towards **understanding and comparing the key components of different GCNs**. Second, comparing with the classical approaches (by Mikhail Belkin and Xiaojin Zhu around 2004-06), despite that we both focus on the linear operation, we take a step forward from another direction, and try to understand and relate various **new GCN models** (e.g., attention-based, topology-based GCNs) in a unified framework. We provide new insights towards state-of-the-art GCN models and inspire to designing better models based on the framework.
>
> 3.	** Training time **
>
> As we have stated in section 4.1, we adopt an iterative algorithm to avoid the computation of Kronecker product and inverse operators. When combining with vanilla GCNs, the training time of our model is similar to GAT and AGNN and faster than APPNP. We have conducted experiments on the training time of baseline models and ours, and we update the results in the supplementary section (Appendix E).
>
> Lastly, we are grateful for your time and we hope we have adequately addressed your concerns. We look forward to additional discussion.
>
> [1] Hoang, NT and T. Maehara. “Revisiting Graph Neural Networks: All We Have is Low-Pass Filters.” ArXiv abs/1905.09550 (2019): n. pag.
> [2] Loukas, Andreas. “What graph neural networks cannot learn: depth vs width.” ArXiv abs/1907.03199 (2020): n. pag.

---

> ### Author Response · Authors · 2020-11-23
> **Update for non-linearity concerns**
>
> To further address the concerns on the non-linearity issue, we update the paper with a newly added section (Sec. 4 in the latest version), which shows that under mild assumptions our main results are still applicable when non-linearity exists. Please refer to Sec.4 of the revised paper for detailed discussion.

---

### Official Review · AnonReviewer2 · 2020-10-28
**Interesting framework but it is not new**

**Rating:** 5
**Confidence:** 5

**Review:**

This paper presents a unified framework for graph convolutional neural networks based on regularized optimization, connecting different  variants of graph neural networks including vanilla, attention-based, and topology-based approaches. The authors also propose  a novel regularization technique to approach the oversmoothing problem in graph convolution. Experiments on the standard settings of node classification on Citeseer, Cora, and Pubmed prove the effectiveness of the proposed regularization techniques.

Overall, this is a very interesting paper, proposing a unified framework for different variants of convolution-based graph neural networks. However, I also have a few concerns:

(1) The proposed framework is mainly designed for GNNs without considering the nonlinear transformation matrix. What if we have to consider the nonlinear transformation? Is the whole framework able to unify different GNNs?

(2) In the case of linear GNNs (without nonlinear transformation matrix), it is actually not surprising formulating GNNs as a regularized optimization problem. Such a regularization framework has already been discussed in the original GCN paper (Kipf et al. 2016).

(3) In the case of linear GNNs, the overall framework is also very similar to the traditional label propagation framework (Zhou et al. Learning with Local and Global Consistency). Could you explain the difference?

(4) The new novel regularization technique seems to be similar to the one proposed in PairNorm (Zhao et al. 2020). Could you also explain the difference?

---

> ### Author Response · Authors · 2020-11-21
> **Response to AnonReviewer2: Part II**
>
> 3. **Differences with traditional label propagation framework **
>
> The main differences between traditional label propagation framework and our work are two-fold. First, our paper proposes a more generalized framework and extends it to **formulating various new GCN models** (e.g., attention-based GCNs, topology-based GCNs). This provides a novel perspective to understand and relate different GCNs. Second, **novel regularizes** can be designed based on our framework and inspire new insights to designing better GCN models.
>
> 4.	**Connections between the proposed regularizer and PairNorm **
>
> The original paper of PairNorm introduces two metrics, which are referred to as **node-wise** and **feature-wise** oversmoothing (eq (3) & (4) in their paper), respectively. However, the PairNorm algorithm fully relies on the first metric, while our regularizer is inspired by the second. As these two metrics are essentially different, our approach differs from PairNorm substantially.
>
> Specifically, PairNorm is a **normalization technique** which designs a normalization layer to prevent unconnected **nodes** from getting closer. By comparison, ours is a **regularization technique** which introduces a regularizer to the objective function to maximize the distance between **features**. Moreover, PairNorm can be viewed as an individual layer, while our regularizer is incorporated in graph filters. Therefore, the two approaches tackle the oversmoothing problem from two different perspectives with different techniques, and can be viewed as complementary methods.
>
> In addition, we find that feature-wise distance is an effective metric for quantifying oversmoothing. As we have stated in section 4.1, features tend to fall into the same subspace spanned by the dominated eigenvectors of the normalized adjacency matrix. Similar observations are also found in [3][4], showing that the distance between features (rather than nodes) shrink to 0 with sufficient number of layers, which is the leading cause of oversmoothing.
>
> Empirically, PairNorm is more suitable when features are missing in the graph, where feature-wise oversmoothing is naturally alleviated. In comparison, our approach achieves consistent improvements in general settings when implemented on several SOTA GCNs.
>
> Lastly, we are grateful for your time and we hope we have adequately addressed your concerns. We look forward to additional discussion.
>
> [1] Hoang, NT and T. Maehara. “Revisiting Graph Neural Networks: All We Have is Low-Pass Filters.” ArXiv abs/1905.09550 (2019): n. pag.
>
> [2] Loukas, Andreas. “What graph neural networks cannot learn: depth vs width.” ArXiv abs/1907.03199 (2020): n. pag.
>
> [3] Klicpera, Johannes et al. “Predict then Propagate: Graph Neural Networks meet Personalized PageRank.” ICLR (2019).
>
> [4] Li, Qimai et al. “Deeper Insights into Graph Convolutional Networks for Semi-Supervised Learning.” ArXiv abs/1801.07606 (2018): n. pag.

---

> ### Author Response · Authors · 2020-11-21
> **Response to AnonReviewer2: Part I**
>
> We would first like to express our appreciation for your time and insightful comments. Please find our response to your concerns in the following:
>
> 1.	** Non-linear activations **
>
> Thanks for the valuable suggestion. Nonlinear activation is indeed important for deep neural networks. Unfortunately, nonlinearity also causes substantial difficulty for theoretical analysis. Many existing works on analyzing GCNs [1][2] only explore the properties of linear operations but still provide thoughtful insights. In our paper, we follow these researches and start with linear operations to understand and relate different GCN models.
> Despite the lack of nonlinear activations, we believe that our framework is still valuable for the following reasons:
>
> First, the key ingredient that differentiates many popular graph neural networks (like GCN, PPNP, and GDC) is the way of designing linear filters, while their nonlinear activation layers are identical (e.g., relu). To analyze the relations and differences among these approaches, it might be sufficient to consider only the linear operations. By formulating various GCNs from the optimization perspective, insights towards understanding and comparing the key components of different GCNs can still be provided.
>
> Second, recent work (e.g., SGC) has shown that even without nonlinear transformations, GCN can still yield competitive results.
>
> 2.	**Differences with regularization framework in vanilla-GCN **
>
> The original GCN paper interprets the graph-based semi-supervised learning problem as adopting a graph Laplacian regularization term with supervised loss. Although the graph Laplacian regularization term is utilized in both frameworks, there still exist several differences:
>
> First, the framework formulated in GCN is a **learning-based** framework driven by supervised signals and back propagation. Therefore, interpretability of the model remains questionable and few insights are discussed in the GCN paper. By comparison, our proposed framework interprets graph filters as closed-form solutions (partly approximated) of **optimization problems**, and shows the property of the filters as various regularizers explicitly.
>
> Second, our proposed framework has better **flexibility** and **potentiality**. Compared to GCN, our unified framework can formulate a variety of GCN-based models and build connections among GCN variants. Meanwhile, expert-designed modules can be implemented based on the unified framework and inspire new GCN models.

---

> ### Author Response · Authors · 2020-11-23
> **Update for non-linearity concerns**
>
> To further address the concerns on the non-linearity issue, we update the paper with a newly added section (Sec. 4 in the latest version), which shows that under mild assumptions our main results are still applicable when non-linearity exists. Please refer to Sec.4 of the revised paper for detailed discussion.

---

### Official Review · AnonReviewer3 · 2020-10-29
**Graph Convolution in a Unified Quandratic Optimization**

**Rating:** 6
**Confidence:** 3

**Review:**

This paper unifies several variants of the graph convolutional networks (GCNs) into a regularized quadratic optimization framework. Basically, the function to be optimized considers both to preserve node information and to perform graph Laplacian regularization, whose optimal solution gives a convolutional layer.

The unification is given by equations (3) and (18) and elaborated in section 3, which includes several methods including GCN, graph attentions, residual connections, concatenation, etc. This is not surprising: as a GCN layer (without activation) is a linear transformation, surely it is the optimum of a quadratic function. Broadly, any linear layer can be trivially formulated as a quadratic optimization problem. Still, I appreciate the authors' delicate work on unifying these diverse methods from an optimization perspective, which is useful and could lead to new methods.

From a technical perspective, the main novelty is that the authors further extend this framework by adding another feature variance term, so that the learned features have a certain variance. This is similar to the idea of batch normalization. This is reasonable because GCN tends to correlate the learned features with the graph Laplacian embedding (the optimal solution of the 2nd term in the authors' framework).

This is interesting but empirical. I would like to see how this additional regularization can be equivalent to transforming the original graph with some formal arguments. Unfortunately, this technic is mainly introduced as a heuristic and more detailed analysis is missing.

As in any regularization framework, there is an additional parameter involved that is the regularization strength (\alpha_3 in 21). Therefore the performance improvement is not surprising as the model is "enlarged". In the experiments or supplementary material, there should be a sensitivity study of this parameter.

On three citation graphs (that are commonly used to evaluate graph neural networks) and semi-supervised node classification tasks, the authors showed that the regularizer can bring marginal performance improvement.

Regarding Clarity, there are some typos in several places and rarely used phrases.

Overall, I don't feel excited after reading the article (although the contents are useful), as a large part of this work is on summarizing existing literature.  The "new bit" is mainly on the additional regularization term that is introduced as a heuristic.

Based on the novelty, a more proper venue for publishing this work could be relevant journals. Overall this submission presents a borderline case and I recommend weak acceptance.

As a minor comment: Equation (21) why not set \alpha_1=1?

----
After rebuttal:

Novelty: my assessment remains the same. It is not non-trivial enough to combine several linear operators into a unified optimization framework. Although the unification is useful, it is not a major novelty.

Thank you for the additional experiments on testing the hyper-parameter. As you mentioned instability, it is worth to have some toy example to demonstrate the instability and study the cause of such instability and show how to avoid such instability using the proposed regularizer. Clearly (19) is bounded. When \alpha_3 is large enough, the solution will be trivial.

Regarding non-linearity: the authors' framework is for unifying a graph convolution operator (that is one layer in a graph neural network). Nonlinear activation is another operator. This is not a major problem from my perspective.

Overall, I think this work has some value (although the novelty is not strong) and still recommend weak acceptance.

---

> ### Author Response · Authors · 2020-11-21
> **Reponse to AnonReviewer3**
>
> We would first like to express our appreciation for your time and insightful comments. Please find our response to your concerns in the following:
> 1.	**Novelty of the paper **
>
> First, we consider the main contribution of our paper is formulating different GCNs in a unified framework. As far as we are concerned, this is the first work trying to **build connections among various GCN-based models**. Due to the success of GCN, numerous variants have been proposed from feature learning, architecture designing and many other directions. Several models are proposed from different angles while end up with similar formulations, e.g., GDC, APPNP and TAGCN. However, the similarities and differences are rarely investigated. Therefore, by establishing a unified framework, we show different modules with regularizers explicitly and relate GCN-based modules from a new perspective.
>
> Then, based on the unified framework, we additionally propose a novel regularizer to tackle oversmoothing. This approach no only achieves better performance, but also shows the feasibility of designing new regularizers based on the framework.
>
> 2.	**Detailed analysis on transforming the original graph **
>
> We have explored the equivalency of graph filters and transformations on the original graph before. However, the analysis is somewhat simple and straightforward, thus we didn’t put it in our paper. For example, the attention-based filters can be interpreted as transferring the hard adjacency matrix to soft adjacency matrix with regard to the attention weights. The topology-based filters can be interpreted as transferring the adjacency matrix to a weighted sum of multiple orders of the original adjacency matrix. In this paper, we are more focused on the effect of various modules from the graph signal perspective, thus we omit the analysis towards the transformation over the graph. Thanks for the valuable suggestion, and we are happy to provide a detailed analysis in the supplementary section in case readers are interested.
>
> 3.	**Sensitive study on the additional hyper-parameter **
>
> We appreciate your suggestions and we have conducted experiments. We update additional results in the supplementary section (Appendix F).
>
> 4.	**The minor comment **
>
> Thanks for pointing out and we are sorry for the confusion. By keeping the \alpha_1, we want to keep the form of (21) the same as the unified framework defined in (18).
>
> Lastly, we are grateful for your time and we hope we have adequately addressed your concerns. We look forward to additional discussion.

---

### Decision · Program_Chairs · 2021-01-07
**Final Decision**

**Decision:**

Reject

**Comment:**

Four reviewers have reviewed and discussed this submission. After rebuttal, two reviewers felt the paper is below acceptance threshold. Firstly, Rev. 1 and Rev. 2 were somewhat disappointed in the lack of analysis regarding non-linearities despite authors suggested this was resolved in the revised manuscript, e.g. Rev. 2 argued the paper without such an analysis is too similar to existing 'linear' models, e.g. APPNP, SGC, and so on. While Rev. 3 was mildly positive about the paper, they also noted that combining several linear operators is somewhat trivial. Overall, all reviewers felt there is some novelty in the proposed regularization term but also felt that contributions of the paper could have been stronger. While AC sympathizes with this submission and hopes that authors can improve this work, in its current form it appears marginally below the acceptance threshold.